# Relationship Between Vitamin D Levels with In-Hospital Complications and Morphofunctional Recovery in a Cohort of Patients After Severe COVID-19 Across Different Obesity Phenotypes

**DOI:** 10.3390/nu17010110

**Published:** 2024-12-30

**Authors:** Víctor J. Simón-Frapolli, Ángel López-Montalbán, Isabel M. Vegas-Aguilar, Marta Generoso-Piñar, Rocío Fernández-Jiménez, Isabel M. Cornejo-Pareja, Ana M. Sánchez-García, Pilar Martínez-López, Pilar Nuevo-Ortega, Carmen Reina-Artacho, María A. Estecha-Foncea, Adela M. Gómez-González, María Belén González-Jiménez, Elma Avanesi-Molina, Francisco J. Tinahones-Madueño, José Manuel García-Almeida

**Affiliations:** 1Department of Endocrinology and Nutrition, Virgen de la Victoria Hospital University Hospital, 29010 Málaga, Spain; victorsimonfrapolli.med@gmail.com (V.J.S.-F.); angel.lmont@gmail.com (Á.L.-M.); martag1996@gmail.com (M.G.-P.); isabelmaria.cornejo@gmail.com (I.M.C.-P.); fjtinahones@uma.es (F.J.T.-M.); 2Facultad de Medicina, University of Málaga, 29010 Málaga, Spain; rociofernandeznutricion@gmail.com; 3Instituto de Investigación Biomédica de Málaga (IBIMA), Plataforma BIONAND, Virgen de la Victoria University Hospital, 29010 Málaga, Spain; isabel.mva13@gmail.com (I.M.V.-A.); anamsgar25@gmail.com (A.M.S.-G.); pimart7@yahoo.es (P.M.-L.); pilarnuevoortega@gmail.com (P.N.-O.); careinart@yahoo.es (C.R.-A.); mae404@hotmail.com (M.A.E.-F.); adelareha@gmail.com (A.M.G.-G.); belen.gonzalez.j@gmail.com (M.B.G.-J.); elma.avanesi.sspa@juntadeandalucia.es (E.A.-M.); 4Department of Critical Care, Virgen de la Victoria Hospital University Hospital, 29010 Málaga, Spain; 5Hospital Virgen de la Victoria, Instituto de Investigación Biomédica de Málaga, 29010 Málaga, Spain; 6Department of Mental Health, Hospital Virgen de la Victoria, 29010 Málaga, Spain; 7Department of Endocrinology and Nutrition, Hospital Quirónsalud, 29004 Málaga, Spain

**Keywords:** vitamin D, nutrition, musculoskeletal disease, immune response, morphofunctional assessment, obesity, sarcopenic obesity, diet-related disease, postcritical SARS-CoV-2 disease

## Abstract

Background and objectives: the COVID-19 pandemic underscored the necessity of understanding the factors influencing susceptibility and disease severity, as well as a better recovery of functional status, especially in postcritical patients. evidence regarding the efficacy of vitamin D supplementation in reducing the severity of COVID-19 is still insufficient due to the lack of primary robust trial-based data and heterogeneous study designs. the principal aims of our study were to determine the impact of vitamin D deficiency or insufficiency on complications during intensive care unit (icu) stay, as well as its role in muscle mass and strength improvement as well as morphofunctional recovery during a multispecialty 6-month follow-up program based on adapted nutritional support and specific physical rehabilitation. as a secondary objective, we compared the association mentioned above between patients with sarcopenic obesity and non- sarcopenic obesity. methods: this prospective observational study included 94 outpatients postcritical COVID-19. two weeks after hospital discharge, patients were divided into sufficient (≥30 ng/mL), insufficient (20.01–29.99 ng/mL), or deficient (≤20 ng/mL) vitamin D levels. the differences in in-hospital complications and morphofunctional parameters including phase angle (PhA), body cell mass (BCM), handgrip strength (HGS), timed get-up-and-go (UAG), 6 min walk test (6MWT), and proinflammatory biochemical variables were analyzed. Incremental (Δ) changes in these parameters were also analyzed at the end of follow-up according to vitamin D levels and the presence vs. absence of sarcopenic obesity. A multivariate linear regression analysis was performed to detect possible confounding factors in the impact analysis of vitamin D changes on functional recovery in patients with obesity. Results: A total of 36.2% of patients exhibited vitamin D deficiency, 29.8% vitamin D insufficiency, and only 32.9% showed sufficient levels at hospital discharge. A total of 46.8% of patients had obesity, and 36.1% had sarcopenic obesity. Vitamin D deficiency was associated with longer hospital stays (*p* = 0.04), longer ICU stays (*p* = 0.04), more days of invasive mechanical ventilation (IMV) (*p* = 0.04), lower skeletal muscle mass/weight (SMM/w) (*p* = 0.04) and skeletal muscle index (SMI) (*p* = 0.047), higher fat mass percentage (FM%) (*p* = 0.04), C-reactive-protein (CRP) (*p* = 0.04), and glycated hemoglobin (HbA1c) (*p* = 0.03), and better performance in R-HGS (*p* = 0.04), UAG (*p* = 0.03), and 6MWT (*p* = 0.034) when compared with those with normal vitamin D levels. At six months, Δvitamin D significantly correlated with ΔHbA1c (*p* = 0.002) and CRP (*p* = 0.049). Patients with normal vitamin D values showed better recovery of ΔSMI (*p* = 0.046), ΔSMM/w (*p* = 0.04), ΔR-HGS (*p* = 0.04), and ΔUAG (*p* = 0.04) compared to those with abnormal vitamin D levels, and these improvements in ΔR-HGS and ΔUAG were greater in the subgroup of sarcopenic obesity compared than in nonsarcopenic obesity (*p* = 0.04 and *p* = 0.04, respectively). Multivariate regression analysis detected that these results were also attributable to a longer hospital stay and lower ΔCRP in the subgroup of patients with sarcopenic obesity. Conclusions: Vitamin D deficiency was associated with longer hospital stays, longer VMI requirement, worse muscle health, and a higher degree of systemic inflammation. Furthermore, normal vitamin D levels at the end of the follow-up were associated with better morphofunctional recovery in postcritical COVID-19, particularly in patients with sarcopenic obesity partly due to a higher degree of inflammation as a result of a longer hospital stay.

## 1. Introduction

Coronavirus disease 2019 (COVID-19) is an infectious and inflammatory respiratory condition caused by the SARS-CoV-2 virus, first identified in December 2019. The severity of the disease varies widely, ranging from asymptomatic or mild cases to severe manifestations characterized by an exaggerated immune response and elevated inflammatory cytokine production, leading to acute respiratory distress syndrome (ARDS), multiorgan failure, or death. In the ICU, infectious syndromes such as ventilator-associated pneumonia, urinary tract infections, and sepsis are the leading causes of morbidity and mortality, often progressing to septic shock and ARDS with profound hypoxemia. COVID-19 has become a defining example of severe infectious disease due to its ability to provoke a range of life-threatening complications in critically ill patients. Among these, ARDS is particularly common, resulting from severe inflammation and damage to the lungs, producing widespread alveolar damage, profound hypoxemia, and respiratory failure, often requiring mechanical ventilation for survival. Additionally, COVID-19 frequently triggers sepsis, a systemic response to infection that can progress to septic shock and multiorgan failure if not promptly addressed [1,2].

The increased survival of critically ill patients has revealed a set of physical, cognitive, and psychological sequelae known as post-ICU syndrome (PICS). The physical sequelae include muscle weakness, fatigue, and persistent respiratory dysfunction, while cognitive deficits may involve memory loss, attention impairments, and reduced executive function. Psychologically, many patients experience anxiety, depression, or post-traumatic stress disorder. One of the associated issues is malnutrition, which raises the risk of rehospitalization and complications. COVID-19 has significantly increased the prevalence of PICS due to its severe systemic inflammation, prolonged immobility, and the requirement for invasive interventions like mechanical ventilation, with high rates of malnutrition and sarcopenia, functional decline, and metabolic imbalances. These sequelae can persist for months, substantially reducing quality of life. Recovery from PICS often requires extensive rehabilitation, underscoring the importance of multidisciplinary care to improve long-term outcomes and quality of life in ICU survivors. The intersection of severe infection, critical illness, and recovery highlights the need for comprehensive approaches in critical care and posthospital management [3,4,5,6,7].

Vitamin D, historically recognized as a global health marker for its critical role in bone health and calcium regulation, has attracted attention as a key modulator of immune function and as a contributor to better muscle mass as well as strength and clinical outcomes. Over the past decades, research has highlighted its indispensable role in both innate and adaptive immune responses, suggesting that insufficient levels of vitamin D may impair the body’s ability to respond effectively to microbial and viral infections [8,9].

In the context of COVID-19, the pandemic highlighted the need to understand the factors that influence susceptibility and disease progression. Low vitamin D levels in hospitalized patients have been associated with increased disease severity, heightened risk of cytokine storm, immune dysregulation, severe complications, and greater mortality and intensive care requirements. Populations with limited sun exposure, malnutrition, or pre-existing chronic illnesses were particularly vulnerable to vitamin D insufficiency, placing them at greater risk [10,11,12].

Vitamin D status is predominantly assessed by serum 25-hydroxyvitamin D concentrations, the principal biochemical marker. Deficiency is generally defined as levels below 20 ng/mL (50 nmol/L), insufficiency as 20–30 ng/mL (50–75 nmol/L), and sufficiency as levels exceeding 30 ng/mL (75 nmol/L). Toxicity, although rare, is characterized by values above 150 ng/mL (375 nmol/L) [13].

In Europe, vitamin D deficiency is widespread, with prevalence depending on the population and the criteria used. When insufficiency is defined as levels below 20 ng/mL, the annual prevalence can reach 40.4%, increasing during winter and declining in summer. Among hospitalized patients, 15–70% exhibit insufficient serum vitamin D levels (<30 ng/mL) at hospital discharge, and approximately 30–50% of these meet the criteria for severe deficiency (<20 ng/mL) [13,14,15,16].

Moreover, vitamin D deficiency has been strongly linked to metabolic disorders such as type 2 diabetes mellitus and obesity (defined as BMI > 30 kg/m^2^). Evidence suggests that low vitamin D levels exacerbate insulin resistance and chronic low-grade inflammation, which are central to the pathophysiology of type 2 diabetes [17]. In obesity, the high prevalence of vitamin D deficiency is partly due to the sequestration of this fat-soluble vitamin in adipose tissue, reducing its bioavailability [9].

Vitamin D deficiency is particularly pronounced in sarcopenic obesity, defined by excess fat mass along with diminished muscle mass and function. According to the Consensus Statement by the European Society for Clinical Nutrition and Metabolism (ESPEN) and the European Association for the Study of Obesity (EASO), sarcopenic obesity should be diagnosed based on fat mass percentage (FM%) exceeding 30% in men and 40% in women, combined with the low muscle mass criterion. Mild sarcopenia is identified when SMM/w is less than one standard deviation (SD) below the reference population values, while severe sarcopenia is classified as an SMM/w below two SDs. Studies indicate that up to 43.3% of individuals with sarcopenic obesity have vitamin D levels below 20 ng/mL, although this varies with the diagnostic criteria used [18,19].

Clinical trials suggest that vitamin D supplementation, particularly when combined with protein intake and physical exercise, can improve muscle strength and prevent muscle mass loss. These interventions have shown significant benefits in maintaining and enhancing muscle mass and strength, especially in individuals at risk of sarcopenia. Trials lasting over 12 weeks have provided robust evidence of vitamin D’s therapeutic potential, with improvements observed in handgrip strength, lower limb strength, and overall physical performance. These findings support the role of vitamin D supplementation as a strategy to mitigate disease severity and enhance recovery outcomes in patients with sarcopenia. Consequently, vitamin D supplementation has emerged as a potential therapeutic approach to improve muscle mass and functional outcomes during follow-up [20,21,22,23].

However, evidence regarding the efficacy of vitamin D supplementation in reducing the severity of COVID-19 and improvement in muscle health parameters is still insufficient due to the lack of primary robust trial-based data and heterogeneous study designs.

The primary aim of our study was to evaluate whether vitamin D deficiency or insufficiency in our cohort is associated with higher rates of complications or aggressive therapeutic requirements during ICU stays as well as with poorer muscle mass and functional recovery during a six-month follow-up rehabilitation program. As secondary objectives, we aimed to investigate the relationship between serum vitamin D levels and the prevalence of diabetes mellitus, obesity, and sarcopenic obesity within our cohort and as well as to evaluate whether vitamin D levels have a greater impact on muscle recovery in patients with sarcopenic obesity compared to patients without sarcopenic obesity.

## 2. Materials and Methods

### 2.1. Study Design

This was a prospective observational study involving 94 patients admitted to the ICU at Hospital Virgen de la Victoria between April 2020 and June 2022 due to severe COVID-19 pneumonia. During hospitalization, all patients were diagnosed with COVID-19 pneumonia in accordance with the World Health Organization (WHO) Interim Guidelines, showing SARS symptoms and confirmed through nasopharyngeal samples taken at admission using real-time reverse-transcription polymerase chain reaction (RT-PCR) testing. Patients admitted to the ICU were those requiring aggressive treatments, including oxygen support greater than 15 L per minute and PaO_2_/FiO_2_ ratios below 200.

Informed consent was obtained from all participants prior to their involvement in this study. This research adhered to the Declaration of Helsinki and received approval from the Ethics Committee of Hospital Virgen de la Victoria (protocol PI-20-321, September 2021). All patients met the inclusion criteria (previous ICU admission for COVID-19 pneumonia and willingness to participate through signed informed consent), while those meeting the exclusion criteria (e.g., refusal to participate, patients from other hospitals making follow-up challenging, patients residing abroad, or inability to perform measurements. Bioelectrical Impedance Vector Analysis (BIVA) due to amputations, extensive skin lesions, or local hematomas) were not included.

Patients were scheduled for an endocrinology- and nutrition-specific consultation between 14 and 21 days after hospital discharge. Sociodemographic and clinical data were collected, and a comprehensive morphofunctional assessment was performed, as described in the subsequent sections. Patients were then classified into three categories based on serum vitamin D levels in blood test performed after discharge. Given that all patients exhibited some degree of malnutrition according to global subjective valoration (GSV) at the time of evaluation [24], all participants were enrolled in a muscle-specific recovery program and were instructed to follow a Mediterranean pattern oral- protein-enriched diet and to supplement it with two servings of an oral nutritional supplement (ONS) daily. The ONS provided 1.5 kcal/mL, delivering 330 kcal, 20 g of protein, 11 g fat, 37 g of carbohydrates, 1.7 g of fiber, 1.5 g of calcium HMB, and 500 IU of vitamin D per 220 mL serving (Ensure^®^ Plus Advance, Abbott Nutrition, Madrid, Spain), and was combined with motor rehabilitation if necessary over a multidimensional six-month follow-up involving multiple specialties (rehabilitation, pulmonology, mental health, and ICU). In this context, once the patients had been assessed regarding global joint balance and global muscle balance during rehabilitation consultation and if they presented any type of deficit, they were offered a comprehensive rehabilitation program, on-site or at home (Appendix A), depending on the following characteristics: The indications for an on-site rehabilitation program were patients with COVID-19 with a negative PCR test with capacity for independent ambulation, stable, presenting dyspnea of moderate to great effort and/or fatigue with moderate effort, fragile with SPPB < 10, and/or need for O_2_. The indications for a home rehabilitation program were fragile patients with no need for O_2_, and patients who could not attend a face-to-face rehabilitation program.

Additionally, patients with vitamin D deficiency or insufficiency received oral 25-hydroxycholecalciferol supplementation during follow-up, in accordance with clinical practice guideline recommendations: if vitamin D insufficiency, 1500–2000 IU/day until normalization; if vitamin D deficiency, 50,000 IU/week for 8 weeks, until vitamin D levels greater than 30 ng/mL were reached, followed by maintenance with doses of 1500–2000 IU/day [13,25].

Patients were reassessed at the end of the follow-up in a specialized nutrition consultation, which included repeated analytical and morphofunctional evaluation. Finally, differences in baseline characteristics between patients with sarcopenic obesity versus those with nonsarcopenic obesity were analyzed, and incremental changes in morphofunctional parameters were analyzed according to the different vitamin D level categories in these two phenotypic subgroups of obesity. It should be noted that non-sarcopenic obesity was defined by BMI but also by FM% criteria according to ESPEN-EASO consensus [18]. The participant flow chart is shown in Figure 1.

### 2.2. Classification According to Vitamin D Levels

Patients were classified according to their serum vitamin D levels into 3 categories following current guidelines: vitamin D deficiency defined as levels ≤ 20 ng/mL (50 nmol/L), insufficiency as 20.01–29.99 ng/mL (50–75 nmol/L), and sufficiency as levels ≥ 30 ng/mL (75 nmol/L) [13]. At the sixmonths consultation, patients were classified into two vitamin D levels categories: <30 ng/mL vs. ≥30 ng/mL.

### 2.3. Body Composition Analysis

#### Phase Angle by Bioelectrical Impedance Vector Analysis (BIVA)

Whole-body bioelectrical impedance measurements were obtained for all patients with a 50 kHz phase-sensitive impedance analyzer (BIA 101 whole-body bioimpedance vector analyzer (AKERN, Pontassieve, Italy)) using tetrapolar 800 mA wearable electrodes placed on the right hand and foot. All patients waited five minutes in the supine position before obtaining their BI measurements. The body consists of complex circuits composed of resistance (Rz) and reactance (Xc) elements that, when stimulated with an alternating current, experience a frequency-dependent current delay with respect to the voltage flow. Through these raw impedance Rz and Xc values, the PhA can be obtained (PhA = arc tangent (Xc/R) × 180°/π). By definition, PhA is positively associated with tissue reactance (associated with cell mass, integrity, function, and composition) and negatively associated with resistance, which depends mainly on the degree of tissue hydration [26].

Individual standardized phase angle (SPhA) values were determined from the sex- and age-matched reference population value by subtracting the reference PhA value from the patient’s observed PhA and then dividing the result by the respective reference standard deviation (SD) by age and sex.

Bioelectrical parameters were analyzed to estimate body composition, such as FM, FM%, fat-free mass (FFM), FFMI, BCM, body cell mass/height (BCM/height), SMM/w, appendicular skeletal muscle mass (ASMM), ASMMI, skeletal muscle index (SMI), total body water (TBW), extracellular body water (ECW), Na/K ratio, and hydration percentage (TBW/FFM).

The interpretation of the impedance values was performed using BIVA. This technique allows for the evaluation of patients by directly determining the impedance vector, independent of equations, predictive models, or body weight, as required in conventional BIA. This method, developed by Piccoli et al. [27], involves plotting the resistance (Rz) and reactance (Xc), standardized by height, as vector points on the resistance–reactance graph (RXc graph). This graph comprises a plane with tolerance ellipses (50%, 75%, and 95% percentiles) defined by the vector distribution of a healthy reference population. Vectors located outside the 75% tolerance ellipse indicate abnormal tissue impedance. Displacement along the major axis of the tolerance ellipses suggests changes in tissue hydration status, i.e., elongated vectors toward the upper pole indicate dehydration, while shortened vectors toward the lower pole indicate hyperhydration. Displacement along the minor axis reflects variations in cellular mass, with vectors shifted to the left indicating greater cellular mass and a larger phase angle, and vectors shifted to the right indicate reduced cellular mass and a smaller phase angle.

### 2.4. Clinical Variables

The following clinical assessments were determined: age, sex, history of diabetes mellitus, BMI grade, presence of obesity according to different phenotypes (obesity by BMI, obesity by FM%, sarcopenic obesity), and Barthel scale. Data on days in ICU, days in hospital, prone maneuvers required, need for oxygen therapy after discharge, and need for rehabilitation were also collected.

### 2.5. Analytical Variables

We determined the following biochemical data at hospital discharge and after 6 months: vitamin D (ng/mL), HbA1c, albumin (g/dL), prealbumin (mg/dL), and CRP (mg/L).

### 2.6. Statistical Analysis

Statistical analyses of the data were performed with Jamovi (version 1.6.23.0 for Mac, Jamovi, Spain).

Descriptive statistics were used to characterize our cohort of patients in general and according to vitamin D levels categories. Baseline characteristics are expressed as mean ± SD for continuous variables and as proportions for categorical variables. Continuous variables were compared with Student’s *t*-test and ANOVA (adding Tukey’s test), or with the Mann–Whitney and Kruskal–Wallis U tests, according to their distribution. Categorical variables were compared with the chi-squared test (or Fisher’s exact test). The relationship between vitamin D and morphofunctional parameters was also analyzed using Pearson’s or Spearman’s correlation models according to the normal distribution.

Using parameters such as ΔPhA, ΔSPhA, ΔTBW/FFM, ΔFM%, ΔBCM, ΔBCM/h, ΔFFMI, ΔASMM, ΔASMMI, ΔSMM/w, ΔSMI, ΔR-HGS, ΔL-HGS, Δalbumin, Δprealbumin, ΔCRP, ΔHbA1c, and ΔUAG, we evaluated the magnitude of the changes in body composition, as well as functional, and biochemical variables after the six-month interventional follow-up period. Finally, we compared these variables across two different vitamin D categories (<30 ng/mL and ≥30 ng/mL) and between the sarcopenic obesity and nonsarcopenic obesity groups to determine whether significant differences existed, using Student’s *t*-test or the Mann–Whitney test as appropriate.

Finally, to analyze the absolute impact of vitamin D changes on functional recovery in patients with sarcopenic and nonsarcopenic obesity, a multivariate linear regression analysis was performed to detect possible confounding factors.

## 3. Results

### 3.1. Baseline Characteristics of Outpatients Postcritical COVID-19 According to Vitamin D Levels

The mean age was 62 ± 12 y, with a mean BMI of 31 ± 7.1 kg/m^2^. A total of 47% were obese using BMI ≥ 30 kg/m^2^, while 48% were obese according to FM% following ESPEN and EASO consensus. A total of 71 patients (75%) were men. Regarding other comorbidities, 31% were previously diabetic, and 36% had sarcopenic obesity. The mean ICU stay was 25 ± 9.5 days, while hospital stay was 49 ± 32 days. During ICU stay, the mean number of prone maneuvers performed in each patient was 2.5 ± 2.4 and the mean duration of IMV was 28 ± 14 days. A total of 44% of patients required home oxygen after discharge, while 89% were referred to a specific rehabilitation program. A 36% of patients were classified as vitamin D deficiency, a 30% as vitamin D insufficiency and a 33% as vitamin D sufficiency. Functional independence, as measured by the Barthel scale, had a mean score of 92 ± 17.

Patients with lower vitamin D levels showed statistically significant differences in sarcopenic obesity prevalence (47% and 43% in the deficiency and insufficiency groups, respectively, vs. 1.9% in the sufficiency group, *p* = 0.04). The ≤20 ng/dL group had longer hospital stays (85 ± 36 days vs. 33 ± 28 days in the ≥30 ng/mL group, *p* = 0.04). When we performed post hoc analysis with Tukey’s multiple comparisons test, we also found statistically significant differences in ICU length, specifically in the group with vitamin D ≤ 20 ng/mL (32 ± 9.8 days) vs. the 20.01–29.99 ng/mL group (19 ± 9.4, *p* = 0.04) and ≥30 ng/mL group (21 ± 8.7, *p* = 0.045) and differences in age in the ≤20 ng/mL (67 ± 10 y) and 20.01–29.99 ng/mL (67 ± 11 y) groups vs. the ≥30 ng/mL group (57 ± 12 y) with *p* = 0.04 and *p* = 0.04 respectively. We also found statistically significant differences in days of IMV in the ≤20 ng/mL (36 ± 14) group compared to the ≥30 ng/mL (14 ± 11) group, with *p*= 0.04 and 0.048, respectively.

When obesity prevalence was analyzed, whether by BMI ≥ 30 kg/m^2^ or by FM%, no statistically significant differences were found. These results are shown in Table 1.

In our sample, statistically significant differences in vitamin D levels were observed across BMI categories (*p* = 0.04) and FM% groups (*p* = 0.04). Patients with a BMI of between 25 and 30 kg/m^2^ had the highest vitamin D levels (32 ± 8.1 ng/mL), while those with a BMI > 40 kg/m^2^ had the lowest levels (15 ± 5.0 ng/mL). Similarly, individuals with an FM% 20–30% exhibited the highest vitamin D levels (30 ± 7.5 ng/mL), whereas those with an FM% > 40% had the lowest (15 ± 7.9 ng/mL). When we performed the Tukey’s test, we observed statistically significant differences between the group aged 60–70 years (23 ± 8.2) and those aged >70 years (16 ± 6.6), with *p* = 0.04. In contrast, vitamin D levels showed no significant variation across sex (*p* = 0.39) or diabetes mellitus (*p* = 0.69), although notable trends included higher levels in men and those without diabetes mellitus. These results are exposed in Table 2.

### 3.2. BIVA Analysis

Statistically significant differences were observed in SMM/w according to vitamin D levels (*p* = 0.04). The ≥30 ng/mL group had a higher SMM/w ratio (35 ± 7.9) than the ≤20 ng/mL group (30 ± 6.1). Performing Tukey’s multiple comparison test, we found statistically significant differences between the vitamin D ≤ 20 ng/mL and D ≥ 30 ng/mL groups in FM% (35 ± 4.0 vs. 31 ± 5.2) and SMI (8.7 ± 1.9 vs. 9.5 ± 2.7), with *p* = 0.04 and *p* = 0.047, respectively. The rest of the parameters and their values depending on vitamin D levels are presented in Table 3.

### 3.3. Functional Status Assessment

A statistically significant difference was observed in 6MWT according to vitamin D categories (*p* = 0.04). Using Tukey’s multiple comparison test, statistically significant differences were seen in the ≤20 ng/mL vs. ≥30 ng/mL groups, in R-HGS, L-HGS, UAG and 6MWT (*p* = 0.04, *p* = 0.040, *p* = 0.03 and *p* = 0.03, respectively) and between the 20.01–29.99 ng/mL group and ≥30 ng/mL group in R-HGS and L-HGS (*p* = 0.04 and *p* = 0.04, respectively). These results are shown in Table 4.

### 3.4. Biochemical Analysis

In the group with vitamin D ≤ 20 ng/mL, the average values were as follows: albumin 2.9 ± 1.0 mg/dL, prealbumin 21± 4.3 g/dL, CRP 51 ± 46 mg/dL, and HbA1c 6.9 ± 1.2%. In the group with vitamin D levels between 20.01 and 29.99 ng/mL, albumin was 3.1 ± 0.9 mg/dL, prealbumin 24 ± 6.4 g/dL, CRP 27 ± 34 mg/dL, and HbA1c 6.0 ± 0.7%. In the group with vitamin D ≥ 30 ng/mL, albumin was 3.4 ± 1.1 mg/dL, prealbumin 26 ± 5.3 g/dL, CRP 15 ± 16 mg/dL, and HbA1c 5.4 ± 0.8%. CRP levels were significantly higher in the group with vitamin D ≤ 20 ng/mL (51 ± 46) than in the other groups, progressively decreasing with increasing vitamin D levels (*p* = 0.04). Similarly, HbA1c levels were significantly higher in the group with vitamin D ≤ 20 ng/mL (6.9 ± 1.2) and lower in the group with levels ≥ 30 ng/mL (5.4 ± 0.8) (*p* = 0.03). No significant differences were observed in albumin or prealbumin levels across the groups, although there was a tendency to have higher albumin and prealbumin levels in the vitamin D ≥ 30 ng/mL subgroup. These results are shown in Table 5.

### 3.5. Correlation of Vitamin D Levels with Validated Measurement of BIVA Parameters, Functional Status Assessment, Biochemical Parameters, and Complications in Patients Postcritical COVID-19

Vitamin D levels showed a significant negative correlation with CRP (039 r = −0.33, *p* = 0.04) and HbA1c (r = −0.52, *p* = 0.01). Additionally, a significant positive correlation was observed with 6MWT (r = 0.52, *p* = 0.01). The full correlation values are presented in Table 6.

### 3.6. Analysis of Morphofunctional Changes at the End of Follow-Up and Their Relationship with Vitamin D Levels

PhA increased from 4.9 ± 1.1 to 5.3 ± 1.1 (Δ = 0.6 ± 0.6, *p* < 0.001), and SPhA rose from −1.1 ± 1.1 to −0.7 ± 1.1 (Δ = 0.6 ± 0.6, *p* < 0.001). FFM increased from 59 ± 12 kg to 60 ± 11 kg (Δ = 1.3 ± 5.0, *p* < 0.001), while FM% decreased from 33 ± 8.8% to 32 ± 8.7% (Δ = −1.7 ± 4.1, *p* = 0.004). R-HGS improved from 23 ± 11 kg to 27 ± 8.0 kg (Δ = 5.6 ± 7.1, *p* < 0.001), and L-HGS increased from 21 ± 12 kg to 25 ± 10 kg (Δ = 6.1 ± 6.3, *p* < 0.001). Functional performance also improved, with UAG decreasing from 9.4 ± 3.0 s to 7.5 ± 1.9 s (Δ = −1.9 ± 2.2, *p* < 0.001) and 6MWT increasing from 365 ± 57 m to 476 ± 51 m (Δ = 116 ± 52, *p* = 0.006). Albumin rose from 3.1 ± 0.9 mg/dL to 3.9 ± 0.4 mg/dL (Δ = 0.9 ± 0.6, *p* < 0.001), CRP dropped from 27 ± 54 mg/dL to 7.6 ± 9.4 mg/dL (Δ = −24 ± 31, *p* = 0.005), and vitamin D increased from 21 ± 8.9 ng/mL to 32 ± 10 ng/mL (Δ = 14 ± 8.9, *p* < 0.001). The alluvial diagram shown in Figure 2 illustrates the changes in the vitamin D levels from baseline to six months, highlighting an overall improvement, particularly among patients with initially lower levels, as many transitioned to higher categories over time. HbA1c and prealbumin showed no significant changes. These results are shown in the Appendix A.

When we analyzed the results based on vitamin D levels at six months, PhA rose to 5.1 ± 1.2 in the vitamin D ≤ 20 ng/mL group, to 5.2 ± 1.2 in the 20.01–29.99 ng/mL group, and to 5.3 ± 1.1 in the ≥30 ng/mL group (*p* = 0.08). Similarly, SPhA improved from −0.8 ± 1.2 in the ≤20 ng/mL group to −0.7 ± 1.2 in the 20.01–29.99 ng/mL group and −0.4 ± 1.2 in the ≥30 ng/mL group (*p* = 0.08). FM% dropped to 37 ± 7.2% in the ≤20 ng/mL group, to 31 ± 8.3% in the 20.01–29.99 ng/mL group, and to 30 ± 8.7% in the ≥30 ng/mL group (*p* = 0.17). Functional performance, as measured by UAG, improved to 8.3 ± 3.2 s in the ≤20 ng/mL group, to 7.8 ± 1.4 s in the 20.01–29.99 ng/mL group, and to 6.2 ± 1.9 s in the ≥30 ng/mL group (*p* = 0.29). However, there were no statistically significant differences. The rest of parameters at six months are shown in the Appendix A.

When examining the magnitude of changes (Δ) over the follow-up period, rather than focusing only on the final value, statistically significant differences were identified based on two vitamin D categories (<30 ng/mL and ≥30 ng/mL).

The ≥30 ng/mL group showed a greater reduction in ΔTBW/FFM (−1.9 ± 3.1 vs. −0.2± 3.3, *p* = 0.03) and ΔFM% (−2.0 ± 3.4 vs. −1.3 ± 6.6, *p* = 0.046). Additionally, improvements were observed in ΔSMI (0.7 ± 1.6 vs. 0.3 ± 0.8, *p* = 0.046, ΔSMM/w (1.0 ± 2.3 vs. 0.4 ± 2.6, *p* = 0.04), ΔR-HGS (8.2 ± 9.1 vs. 4.7 ± 6.3, *p* = 0.04), and ΔUAG (−2.3 ± 2.9 vs. −1.6 ± 2.4, *p* = 0.04).

Although no statistical significance was observed, the ΔNa/K ratio showed a greater reduction in the ≥30 ng/mL vitamin D group (−0.1 ± 0.2) compared to the <30 ng/mL group (−0.04 ± 0.2, *p* = 0.08). ΔSMM and ΔASMM were higher in the ≥30 ng/mL group (ΔSMM: 1.47 ± 1.5, ΔASMM: 1.28 ± 3.0) compared to the <30 ng/mL group (ΔSMM: 1.1 ± 1.3, ΔASMM: 1.0 ± 1.5), with *p* = 0.10 and *p* = 0.28, respectively. Δalbumin also showed a greater increase in the ≥30 ng/mL group (1.0 ± 0.5 mg/dL) compared to the <30 ng/mL group (0.7 ± 0.7 mg/dL, *p* = 0.09). These results are shown in Table 7.

The correlation analysis revealed that changes in vitamin D levels (Δvitamin D) were correlated with various clinical parameters. A significant negative correlation was observed with changes in HbA1c (r = −0.77, *p* = 0.002). Similarly, a moderate negative correlation was found with hospital stay (r = −0.33, *p* = 0.04), with ΔCRP (r =−0.39, *p* = −0.049) and with ΔUAG (r = −0.34, *p* = −0.09), although the latter was not statistically significant. These results are exposed in Table 8.

To further investigate the impact of vitamin D levels on morphofunctional recovery in patients postcritical COVID19 with different obesity phenotypes, we first studied the baseline characteristics of the patients with sarcopenic and nonsarcopenic obesity. When we compared people with nonsarcopenic obesity vs. sarcopenic obesity, statistically significant differences were found in age (56 ± 13 vs. 64 ± 11, *p* = 0.02), sex (73% men vs. 18%, *p*= 0.02), hospital stay (33 ± 19 vs. 71 ± 51, *p* < 0.001), ICU stay (18 ± 16 vs. 47 ± 35, *p* < 0.001) and CRP (19 ± 29 vs. 59 ± 47, *p* = 0.04). These results are shown in Table 9.

When we analyzed the above results based on the presence of sarcopenic obesity versus nonsarcopenic obesity, in each of the two vitamin D categories, we obtained statistically significant differences for some functional tests. In the vitamin D < 30 ng/mL group, individuals with sarcopenic obesity showed a greater ΔR-HGS (7.1 ± 5.1 kg) compared to those with nonsarcopenic obesity (2.7 ± 2.5 kg, *p* = 0.048). Similarly, in the vitamin D ≥ 30 ng/mL group, individuals with sarcopenic obesity demonstrated significant improvements in both R-HGS (ΔR-HGS: 8.7 ± 13 kg vs. 2.5 ± 9.8 kg, *p* = 0.04), L-HGS (ΔL-HGS: 9.0 ± 9.8 kg vs. 0.5 ± 3.5 kg, *p* = 0.002) and AUG (−2.2 ± 1.1 s vs. −1.1 ± 1.1 s, *p*= 0.04), when compared with patients with nonsarcopenic obesity. Although no statistical significance was observed, the Na/K ratio showed a more pronounced reduction in those with sarcopenic obesity, particularly among those with higher vitamin D levels (ΔNa/K ratio: −0.1 ± 0.2 vs. −0.1 ± 0.1, *p* = 0.34). Additionally, in the vitamin D ≥ 30 ng/mL group, individuals with sarcopenic obesity tended to have greater increases in the skeletal muscle mass index (ΔSMI: 1.6 ± 0.9 vs. 1.2 ± 1.1, *p* = 0.70) compared to those with nonsarcopenic obesity. These results are shown in Table 10.

### 3.7. Multivariate Lineal Regression Analysis of Predictors of ΔR-HGS in Patients with Obesity

Linear regression analysis was performed to evaluate the predictive power of various factors affecting ΔR-HGS in patients with obesity and to determine possible confounding factors. The statistically significant variables associated with the ΔR-HGS were ΔvitaminD (estimated coefficient 0.30930, *p* = 0.025), hospital stay (estimated coefficient 0.0936, *p* = 0.048), and ΔCRP (estimated coefficient −0.2105, *p* = 0.031). These results are fully shown in Table 11.

## 4. Discussion

The principal findings of our study demonstrate a strong association between low levels of vitamin D, especially with values ≤ 20 ng/mL, and a higher prevalence of obesity and sarcopenic obesity, as well as their potential impact on the severity of COVID-19. This severity is reflected in prolonged hospital stays, extended ICU stays, increased VMI requirements, and elevated inflammatory markers such as CRP and HbA1c. Furthermore, vitamin D levels positively influenced the morphofunctional recovery of patients with PICS, leading to greater improvements in muscle mass and strength and better performance on functional tests, particularly among patients with sarcopenic obesity. To the best of our knowledge, no prior studies explored the impact of vitamin D on bioelectric parameters and functional tests in a prospective manner in patients affected by PICS after severe COVID-19 disease. Therefore, this work offers a novel approach.

Vitamin D also played a critical role during morphofunctional recovery programs in postcritical patients after COVID-19 pneumonia. Special emphasis should be placed on patients with sarcopenic obesity, as vitamin D levels greater than 30 ng/mL were associated with more favorable changes in muscle health parameters when compared to non-sarcopenic obese patients. However, there has traditionally been no consensus on the diagnostic criteria for sarcopenic obesity, which made it difficult to detect these patients and limited the scientific evidence when establishing therapeutic recommendations.

Vitamin D’s role in reducing systemic inflammation and maintaining metabolic homeostasis is critical for the recovery of patients from severe illnesses such as COVID-19. Vitamin D modulates inflammation, particularly in patients with underlying inflammatory or infectious conditions, and supports glycemic control in patients with diabetes. Its anti-inflammatory effects are mediated through the downregulation of IL-6 and TNF-α and the upregulation of IL-10, thereby mitigating chronic inflammation that contributes to insulin resistance and β-cell dysfunction. Moreover, vitamin D improves HbA1c by enhancing insulin sensitivity and secretion through its interaction with the vitamin D receptor (VDR) on pancreatic β cells. In contrast, systemic inflammation can reduce vitamin D levels through mechanisms such as increased 24-hydroxylase activity, impaired synthesis, reduced absorption, and sequestration in adipose tissue [28,29,30,31]. In fact, there is evidence to support that vitamin D levels drop in acute respiratory tract infections [32].

Our findings of extended hospital stays, prolonged ICU admission, and increased VMI requirements in patients with vitamin D deficiency align with studies that reported that patients with COVID-19 supplemented with vitamin D experienced better clinical recovery, shorter hospital and ICU stays, and reduced mortality during ICU admission, highlighting the overall positive impact of vitamin D on inflammation and nutritional parameters [12,33,34].

In our cohort, 36% of the patients exhibited vitamin D deficiency, 30% had vitamin D insufficiency, and only 33% had sufficient levels of vitamin D. Thus, 66% of patients presented with abnormal vitamin D levels, consistent with findings in the existing literature regarding hospitalized patients [13,15,16]. Our sample had a high prevalence of obesity (47% by BMI and 48% by FM%), which likely contributed to vitamin D deficiency due to sequestration in adipose tissue, reducing its bioavailability [9]. Interestingly, when analyzing vitamin D values based on obesity degree, a J-shaped curve was observed. Patients with a BMI < 25% and a FM < 20% (categories that correspond to those who are potentially the most malnourished) showed slightly lower vitamin D levels. These levels peaked in the BMI 25–30% and FM 20–30% groups before progressively decreasing as BMI and FM% increased. The prevalence of sarcopenic obesity was 36%, being significantly higher in patients with vitamin D deficiency compared to those with normal vitamin D levels. Notably, only 2% of patients with sarcopenic obesity had normal vitamin D levels, compared to 32% and 32% in those with obesity defined by FM% and BMI, respectively, differences that were also noted in prior studies [19]. However, no differences in vitamin D levels were observed between patients with and without diabetes, contrasting the literature suggesting greater deficiencies among individuals with diabetes [35].

When assessing bioimpedance parameters according to vitamin D level, significant differences were noted in SMM/w and, to a lesser extent, in SMI, when comparing the ≤20 ng/mL group vs. ≥30 ng/mL group. No statistical differences were observed in the other muscle mass parameters, including FFM, FFMI, BCM, BCM/h, ASMM, ASMMI, or ASMM. A possible explanation is that patients with obesity generally exhibit higher absolute muscle mass. They also show a greater FM%, facilitating the sequestration of vitamin D [9]. As the ≥30 ng/mL vitamin D group had a lower obesity rate, using parameters like SMM/w that adjust muscle mass by total weight provided a more discriminatory measure, highlighting significant differences.

Regarding muscle strength and functional performance, statistically significant differences were found in R-HGS, L-HGS, UAG, and 6MWT, with superior results observed among patients with sufficient vitamin D levels. Additionally, inflammatory markers such as FM%, TBW/FFM, CRP, HbA1c, and Na/K ratio were evaluated. FM%, HbA1c, and CRP showed statistically significant differences, with higher values in patients with lower vitamin D levels [28].

The incremental changes (Δ) in morphofunctional parameters over six months of a multidimensional follow-up recovery program, including specific nutritional support and a structured physical exercise program, demonstrated generalized improvements in variables such as ΔPhA, ΔSPhA, ΔBCM, and ΔBCM/h, as well as in functional status. These data support the results described in previous publications on the approach to PICS [36]. The incremental change in vitamin D was also remarkable, both due to specific supplementation during follow-up and probably to cure of respiratory infection with COVID-19 with improvements in the proinflammatory environment [32]. The impact of seasonal changes on the increase in vitamin D levels was not evaluated, although months with more sunlight could have had a positive influence.

It is important to note that when analyzing the final values (at six months) of the morphofunctional parameters according to vitamin D levels, no statistically significant differences were found, because all patients tended to improve following the nutritional intervention. However, when analyzing the incremental changes (Δ) in these parameters, significant differences were observed in muscle strength measured by ΔHGS, muscle function measured by ΔUAG, and muscle mass measured by ΔSMI and ΔSMM/w, resulting in greater improvements in patients with normal levels of vitamin D at the end of follow-up. This improvement may have been due not only to the normalization of vitamin D levels but also to a synergistic effect of the specific rehabilitation program and the adapted nutritional support. However, 100% of the patients received ONS, and 89% completed the rehabilitation program, so the differential impact on each category of vitamin D level was practically nil, although we could not exactly determine the absolute impact of each of these variables on the improvement in vitamin D levels in our entire cohort.

When examining incremental changes based on patients with sarcopenic versus nonsarcopenic obesity, we eliminated obesity as a confounder in the vitamin D analysis, which allowed us to analyze sarcopenia independently of FM. Sufficient vitamin D levels at follow-up were associated with greater recovery of muscle strength and function in the sarcopenic obesity group, evidenced by improved ΔR-HGS, ΔL-HGS, and ΔUAG. Notably, improvement in ΔAUG was observed exclusively in the sarcopenic obesity subgroup. These results are difficult to compare with the previous literature due to the lack of homogeneity in applying the diagnostic criteria for sarcopenic obesity and given the absence of studies. Furthermore, in order to analyze the possible confounding factors in the analysis of the impact of vitamin D levels on ΔR-HGS in patients with obesity, a multivariate regression analysis was performed. In our model, an R of 0.6780 represented a moderate to strong correlation between the independent variables and the dependent variable, suggesting a substantial linear relationship in our model. The results reinforced the finding of the effect of ΔvitaminD being independent of the other variables analyzed and highlighted its role as a key determinant in muscle strength recovery, even after adjusting for important factors such as ΔCRP and hospital stay, which also affected ΔR-HGS in a statistically significant way. However, these data suggested that the functional improvement found in the patient subgroups of sarcopenic obesity versus nonsarcopenic obesity may have been due not only to vitamin D level normalization but also to an improvement in proinflammatory markers that were elevated at baseline after prolonged hospitalization.

Despite the significant vitamin D deficiency observed in hospitalized patients, especially those with sarcopenic obesity, levels were not restored to normal in 43% during follow-up (8% showed deficient levels, and 35% showed insufficient levels), highlighting the need for intensive monitoring and treatment, particularly for patients with obesity requiring higher supplementation doses [13].

## 5. Limitations

This study has several limitations. Firstly, a survival bias was evident as the analysis included only patients who survived ICU admission, excluding those who died early during their ICU stay. Additionally, there was a notable predominance of male participants in the sample, potentially limiting the generalizability of the findings across the sexes.

Another limitation was the inability to perform 6MWT during the six-month follow-up visit in some cases, primarily due to time constraints and a lack of the necessary infrastructure. Consequently, we were unable to analyze the Δ6MWT according to vitamin D levels, which could have been a valuable variable for assessing functional improvements in these patients.

Although the initial sample size was 94, the subgroup analysis of patients with sarcopenic obesity versus nonsarcopenic obesity resulted in a significantly smaller sample size (n = 50) with fewer patients in each subgroup, notably limiting the statistical power and significance of these analyses. However, we still achieved statistical significance for some parameters.

Moreover, the analysis of morphofunctional parameter changes according to vitamin D levels was influenced by potential confounding variables not studied in the multivariate regression analysis, such as baseline nutritional status, physical activity level, specific nutritional support, and underlying comorbidities during follow-up. In the analysis of complications and aggressive therapy requirements in the ICU, some additional potential confounding factors were baseline malnutrition status and pre-existing comorbidities. Although most of these potential confounding factors had a similar distribution in each category of vitamin D values, these factors may have impacted the observed outcomes and complicated the interpretation of the results.

Regarding the in-hospital complications analysis, another important limitation is that our results demonstrate association rather than causality, given that vitamin D levels were assessed after admission and not prior to it. Future research using randomized controlled trials (RCTs) would provide more robust evidence regarding the effectiveness of vitamin D supplementation in improving the recovery outcomes for patients with COVID-19 and the optimal strategies for achieving it in this vulnerable patient population. Finally, extending the follow-up period beyond six months would help assess whether the benefits of normal vitamin D levels persist over time and whether long-term supplementation is necessary to maintain these benefits. Furthermore, mechanistic studies examining the underlying processes through which vitamin D influences inflammation, muscle mass, and strength could provide deeper insights into how this micronutrient contributes to recovery in patients after critical COVID-19.

## 6. Conclusions

Vitamin D level deficiency was associated with more complications and requiring aggressive therapies during ICU admission and with the improvement in clinical and morphofunctional outcomes in patients postcritical COVID-19, particularly those with sarcopenic obesity, partly due to a higher degree of inflammation as a result of a longer hospital stay. Therefore, it would be appropriate to measure vitamin D levels and to treat for them, if necessary, during the recovery from post-ICU syndrome after severe COVID-19 disease. While our findings are partly consistent with the existing literature, addressing limitations through larger, controlled studies and prolonged monitoring could yield more definitive insights.

## Figures and Tables

**Figure 1 nutrients-17-00110-f001:**
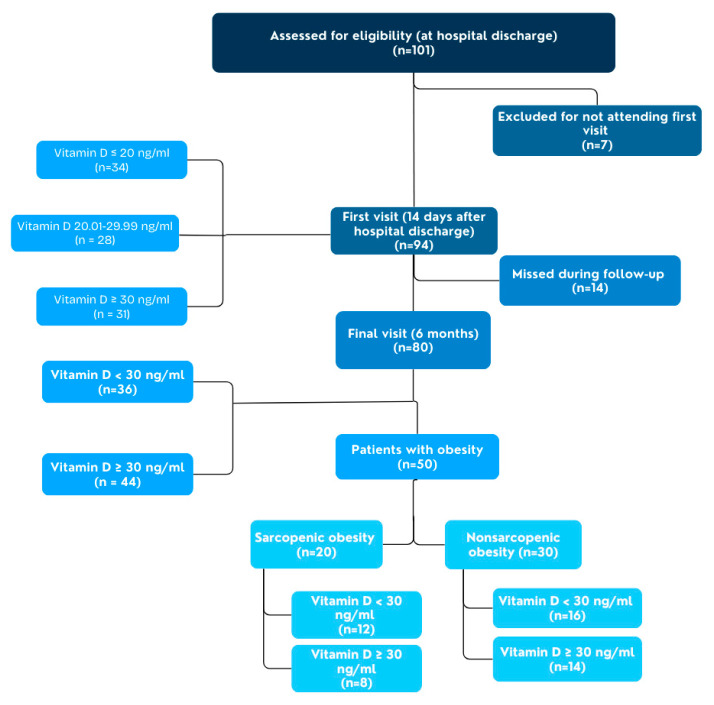
Participant flow chart.

**Figure 2 nutrients-17-00110-f002:**
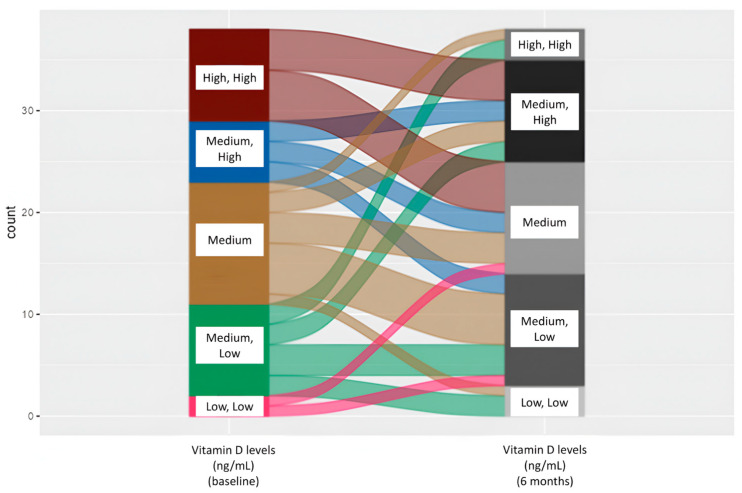
Alluvial diagram of the changes in vitamin D levels from baseline to six months.

**Table 1 nutrients-17-00110-t001:** Baseline characteristics according to vitamin D level.

	All	Vitamin D ≤ 20 ng/mL	Vitamin D 20.01–29.99 ng/mL	Vitamin D ≥ 30 ng/mL	*p* Value
	N = 94	N = 34	N = 28	N = 31	
Demographic variable					
Age (years)	62 ± 12	67 ± 10 ^b^	67 ± 11 ^c^	57 ± 12 ^b,c^	0.32 ^b,c^
Male (%)	71 (75)	25 (73)	23 (82)	23 (74)	0.86
Diabetes mellitus (%)	29 (31)	10 (29)	12 (43)	8 (26)	0.53
BMI (kg/m^2^)	31 ± 7.1	34 ± 8.6	30 ± 7.1	29 ± 6.2	0.20
Obesity by BMI ≥ 30 kg/m^2^ (%)	44 (47)	17 (35)	17 (61)	10 (32)	0.23
Grade 1 (30–34.99 kg/m^2^)	24 (25)	10 (31)	10 (36)	4 (13)	
Grade 2 (35–39.99 kg/m^2^)	10 (11)	2 (6)	5 (18)	3 (10)	
Grade 3 (40–49.99 kg/m^2^)	10 (11)	5 (15)	2 (7)	3 (10)	
Obesity by FM ≥ 30% (men) or ≥40% (women)	45 (48)	20 (59)	15 (53)	10 (32)	0.10
Sarcopenic obesity	34 (36)	16 (47) ^b^	12 (43) ^c^	6 (2) ^b,c^	0.04 *
Complications					
ICU stay (days)	25 ± 9.5	33 ± 9.8 ^a,b^	19 ± 9.4 ^a^	21 ± 8.7 ^b^	0.09 ^a^
Hospital stay (days)	49 ± 32	85 ± 36 ^b^	50 ± 24	33 ± 28 ^b^	0.04 *
Invasive mechanical ventilation (days)	28 ± 14	36 ± 14 ^b^	23 ± 14	14 ± 11 ^b^	0.84 ^b^
Prone maneuvres (n)	2.5 ± 2.4	2.6 ± 1.3	2.6 ± 1.3	2.4 ± 1.2	0.34
Home oxygen therapy after hospital discharge (%)	41 (44)	7 (56)	19 (57)	15 (48)	0.40
Rehabilitation (%)	84 (89)	32 (94)	25 (89)	26 (84)	0.15
Barthel scale	92 ± 7.4	96 ± 8.5	90 ± 11	98 ± 5.6	0.38

Data are expressed as mean ± standard deviations. Groups were divided according to vitamin D levels. Asterisk indicates significant difference between groups, according to ANOVA (or Kruskal–Wallis U test) and Tukey’s multiple comparison test when required (* *p* < 0.05). ^a^ Statistically significant differences between vitamin D ≤ 20 ng/mL and vitamin D 20.01–29.99 ng/mL; ^b^ statistically significant differences between vitamin D ≤ 20 ng/mL and vitamin D ≥ 30 ng/mL; ^c^ statistically significant differences between vitamin D 20.01–29.99 ng/mL and vitamin D ≥ 30 ng/mL. Chi-squared test (or Fisher’s exact test) was used for variables expressed as percentage (* *p* < 0.05). Abbreviations: BMI: body mass index; FM: fat mass; ICU: intensive care unit.

**Table 2 nutrients-17-00110-t002:** Vitamin D levels according to baseline characteristics of study population.

	N	Vitamin D (ng/mL)	*p* Value
Age (years)			0.25 ^f^
<50	12	22 ± 6.6	
50–60	16	17 ± 8.9	
60–70	37	23 ± 8.2 ^f^	
>70	29	16 ± 6.6 ^f^	
Sex			0.39
Female	27	19 ± 5.1	
Male	67	22 ± 5.0	
Diabetes Mellitus			
Yes	38	21 ± 9.7	0.693
No	56	18 ± 9.5	
BMI (kg/m^2^)			0.04 *
<25	22	22 ± 6.7	
25–30	28	32 ± 8.1 ^f,g^	
30–35	21	20 ± 11	
35–40	12	19 ± 7.2 ^f^	
>40	11	15 ± 5.0 ^g^	
FM%			0.04 *
<20%	6	30 ± 7.5	
20–30%	24	31 ± 7.0 ^d,e^	
30–40	50	20 ± 8.4 ^d^	
>40%	14	15 ± 7.9 ^e^	

Data are expressed as mean ± standard deviations or percentage. Groups were divided according to baseline characteristics. Asterisk indicates significant difference between groups, according to Student’s *t*-test or ANOVA test (* *p* < 0.05). ^d^ statistically significant differences between 50–60 y and 60–70 y, BMI < 25 kg/m^2^ and BMI > 40 kg/m^2^, FM 20–30% and FM 30–40% (*p* < 0.05); ^e^ statistically significant differences between 50–60 y and >70 y, BMI 25–30 kg/m^2^ and BMI 30–35 kg/m^2^, FM 20–30% and FM > 40% (*p* < 0.05); ^f^ statistically significant differences between 60–70 y and >70 y, BMI 25–30 kg/m^2^ and BMI 35–40 kg/m^2^, FM 30–40% and FM > 40% (*p* < 0.05). ^g^ Statistically significant differences between BMI 25–30 kg/m^2^ and BMI > 40 kg/m^2^ (*p* < 0.05).

**Table 3 nutrients-17-00110-t003:** Bioelectrical impedance variables according to vitamin D levels.

	All	Vitamin D ≤ 20 ng/mL	Vitamin D 20.01–29.99 ng/mL	Vitamin D ≥ 30 ng/mL	*p* Value
	N = 94	N = 34	N = 28	N = 31	
PhA	4.9 ± 1.1	4.3 ± 1.0	4.8 ± 1.1	4.6 ± 1.1	0.39
SPhA	−1.1 ± 1.1	−1.4 ± 0.3	−1.0 ± 0.4	−1.3 ± 0.3	0.31
TBW/FFM (%)	77 ± 4.1	77 ± 4.6	76 ± 4.1	76 ± 4.2	0.36
FM (%)	34 ± 8.8	35 ± 4.0 ^b^	34 ± 5.1	31 ± 5.2 ^b^	0.09 ^b^
FFM (kg)	57 ± 12.0	54 ± 6.6	55 ± 5.7	57 ± 5.9	0.35
FFMI (kg/m^2^)	20 ± 3.2	19 ± 3.8	19 ± 3.6	20 ± 3.6	0.31
BCM (kg)	26 ± 8.9	23 ± 4.7	27 ± 5.1	27 ± 4.6	0.20
BCM/h (kg/m)	16 ± 4.8	14 ± 3.0	15 ± 3.5	16 ± 3.3	0.19
Na/K ratio	1.2 ± 0.2	1.3 ± 0.3	1.1 ± 0.3	1.1 ± 1.2	0.31
ASMM (kg)	23 ± 5.9	21 ± 2.1	23 ± 2.4	23 ± 1.9	0.27
ASMMI (kg/m^2^)	7.9 ± 1.6	7.5 ± 1.7	7.7 ± 1.9	8.0 ± 2.1	0.27
SMM (kg)	28 ± 6.9	24 ± 4.3	27 ± 4.4	28 ± 4.7	0.14
SMI (kg/m^2^)	9.7 ± 1.9	8.7 ± 1.9 ^b^	9.3 ± 2.4	9.5 ± 2.7 ^b^	0.13 ^b^
SMM/w	32 ± 5.6	30 ± 6.1 ^b^	33 ± 3.3	35 ± 7.9 ^b^	0.04 *

Data are expressed as mean ± standard deviations. Groups were divided according to vitamin D level. Asterisk indicates significant difference between groups, according to ANOVA (or Kruskal–Wallis U test) and Tukey’s multiple comparison test when required (* *p* < 0.05). Abbreviations: PhA: phase angle; SPhA: standardized phase angle; TBW: total body water; FFM: fat-free mass; BCM: body cell mass; BCM/h: standardized body cell mass; ASMM: appendicular skeletal muscle mass; SMM: skeletal muscle mass; SMM/w: skeletal muscle mass/weight; SMI: skeletal mass index; ASMMI: appendicular skeletal muscle mass index; FM: fat mass. ^b^ statistically significant differences between vitamin D ≤ 20 ng/mL and vitamin D ≥ 30 ng/mL.

**Table 4 nutrients-17-00110-t004:** Functional test variables according to vitamin D levels.

	All	Vitamin D ≤ 20 ng/mL	Vitamin D 20.01–29.99 ng/mL	Vitamin D ≥ 30 ng/mL	*p* Value
	N = 94	N = 34	N = 28	N = 31	
R-HGS (kg)	23 ± 11	20 ± 4.0 ^b^	20 ± 5.1 ^c^	24 ± 4.7 ^b,c^	0.07 ^b^
L-HGS (kg)	21 ± 12	18 ± 3.3 ^b^	19 ± 3.4 ^c^	23 ± 2.3 ^b,c^	0.07 ^b^
UAG (s)	9.4 ± 3.0	11 ± 2.6 ^b^	11 ± 2.1	8.8 ± 2.2 ^b^	0.06 ^b^
6MWT (m)	365 ± 57	321 ± 52	375 ± 54	407 ± 58	0.04 *

Data are expressed as mean ± standard deviations or percentage. Groups were divided according to vitamin D level. Asterisk indicates significant difference between groups, according to ANOVA (or Kruskal–Wallis U test) and Tukey’s multiple comparison test when required (* *p* < 0.05). Abbreviations: R-HGS: right handgrip strength; L-HGS: left handgrip strength; UAG: timed get-up-and-go; 6MWT: 6 min walk test; ^b^ statistically significant differences between vitamin D ≤ 20 ng/mL and vitamin D ≥ 30 ng/mL; ^c^ statistically significant differences between vitamin D 20.01–29.99 ng/mL and vitamin D ≥ 30 ng/mL.

**Table 5 nutrients-17-00110-t005:** Biochemical parameters according to vitamin D level.

	All	Vitamin D ≤ 20 ng/mL	Vitamin D 20.01–29.99 ng/mL	Vitamin D ≥ 30 ng/mL	*p* Value
	N = 94	N = 34	N = 28	N = 31	
Albumin (mg/dL)	3.1 ± 0.9	2.9 ± 1.0	3.1 ± 0.9	3.4 ± 1.1	0.15
Prealbumin (g/dL)	24 ± 5.2	21 ± 4.3	24 ± 6.4	26 ± 5.3	0.35
CRP (mg/dL)	27 ± 54	51 ± 46	27 ± 34	15 ± 16	0.04 *
HbA1c (%)	6.2 ± 1.1	6.9 ± 1.2	6.0 ± 0.7	5.4 ± 0.8	0.03 *

Data are expressed as mean ± standard deviations or percentage. Groups were divided according to vitamin D levels. Asterisk indicates significant difference between groups, according to ANOVA (or Kruskal–Wallis U test) and Tukey’s multiple comparison test when required (* *p* < 0.05). Abbreviations: CRP: C-reactive protein. HbA1c: glycated hemoglobin.

**Table 6 nutrients-17-00110-t006:** Correlation of vitamin D levels with functional status assessment and biochemical parameters in outpatients after critical COVID-19.

		HbA1c	CRP	6MWT
Vitamin D	Pearson’s r *p* value	−0.52 0.01 *	−0.33 0.04 *	0.52 0.01 *

Correlation was analyzed using Pearson’s or Spearman’s correlation models depending on their distribution. Asterisk indicates significant difference between groups (* *p* < 0.05).

**Table 7 nutrients-17-00110-t007:** Changes (Δ) in morphofunctional parameters at six months, according to vitamin D level.

	All	Vitamin D < 30 ng/mL	Vitamin D ≥ 30 ng/mL	*p* Value
	N = 94	N = 36	N = 44	
ΔPhA	0.6 ± 0.6	0.6 ± 0.9	0.7 ± 0.6	0.53
ΔSPhA	0.6 ± 0.6	0.6 ± 0.9	0.7 ± 0.6	0.51
ΔTBW/FFM (%)	−1.1 ± 3.7	−0.2 ± 3.3	−1.9 ± 3.1	0.03 *
ΔFM (%)	−1.7 ± 4.1	−1.3 ± 6.6	−2.0 ± 3.4	0.046 *
ΔFFM (kg)	1.3 ± 5.0	1.2 ± 3.9	1.3 ± 7.0	0.29
ΔFFMI (kg/m^2^)	1.2 ± 1.7	0.9 ± 1.4	1.3 ± 1.4	0.29
ΔBCM (kg)	3.5 ± 4.6	3.4 ± 4.7	3.7 ± 3.5	0.58
ΔBCM/h (kg/m)	2.2 ± 2.6	2.1 ± 3.8	2.3 ± 2.1	0.59
ΔNa/K ratio	−0.1 ± 0.5	−0.04 ± 0.2	−0.1 ± 0.2	0.08
ΔASMM (kg)	1.2 ± 2.1	1.0± 1.5	1.3 ± 3.0	0.28
ΔASMMI (kg/m^2^)	0.3 ± 0.7	0.2 ± 0.3	0.3 ± 0.6	0.34
ΔSMM (kg)	1.3 ± 2.3	1.1 ± 1.3	1.5 ± 1.5	0.11
ΔSMI (kg/m^2^)	0.4 ± 1.2	0.3 ± 0.8	0.7 ± 1.6	0.046 *
ΔSMM/w	0.5 ± 2.5	0.4 ± 2.6	1.0 ± 2.3	0.04 *
ΔR-HGS (kg)	5.6 ± 7.1	4.7 ± 6.3	8.2 ± 9.1	0.04 *
ΔL-HGS (kg)	6.1 ± 6.3	5.6 ± 5.3	8.1 ± 7.2	0.11
ΔUAG (s)	−1.9 ± 2.2	−1.6 ± 2.4	−2.3 ± 2.9	0.04 *
ΔAlbumin (mg/dL)	0.9 ± 0.6	0.7 ± 0.7	1.03 ± 0.5	0.09
ΔPrealbumin (g/dL)	3.0 ± 11	1.4 ± 14	3.8 ± 12	0.31
ΔCRP (mg/dL)	−24 ± 31	−18 ± 64	−35 ± 68	0.38
ΔHbA1c (%)	−0.01 ± 1	−0.01 ± 1	−0.01 ± 0.8	0.51

Data are expressed as mean ± standard deviations. Groups were divided according to vitamin D levels. Abbreviation: PhA: phase angle; SPhA: standardized phase angle; TBW: total body water; FFM: fat-free mass; BCM: body cell mass; BCM/h: standardized body cell mass; BMI: body mass index; FM: fat mass; ASMM: appendicular skeletal muscle mass; SMM: skeletal muscle mass; SMM/w: skeletal muscle mass/weight; SMI: skeletal mass index; ASMMI: appendicular skeletal muscle mass index; CRP: C-reactive protein; HbA1c: glycated hemoglobin. Asterisk indicates significant difference between groups, according to Student’s *t*-test (or Mann–Whitney test) (* *p* < 0.05).

**Table 8 nutrients-17-00110-t008:** Correlation between incremental changes in vitamin D levels with incremental changes in functional tests, hospital and ICU stays, and biochemical parameters in outpatients after critical COVID-19 after a 6-month follow-up.

		ICU Stay	Hospital Days	ΔHbA1c	ΔCRP	ΔUAG	ΔR-HGS
Δvitamin D	Pearson’s r *p* value	−0.29 0.06	−0.33 0.04 *	−0.77 0.002 *	−0.39 0.049 *	−0.34 0.10	0.30 0.12

Correlation was analyzed using Pearson’s or Spearman’s correlation models depending on their distribution. Asterisk indicates significant difference between groups (* *p* < 0.05).

**Table 9 nutrients-17-00110-t009:** Baseline characteristics of those with sarcopenic and nonsarcopenic obesity.

	All	Nonarcopenic Obesity	Sarcopenic Obesity	*p* Value
	N = 94	N = 30	N = 20	
Age (y)	62 ± 12	56 ± 13	64 ± 11	0.02 *
Male (%)	71 (75)	22 (73)	18 (90)	0.02 *
Hospital stay (days)	49 ± 32	33 ± 19	71 ± 51	<0.001 *
ICU stay (days)	25 ± 9.5	18 ± 16	47 ± 35	<0.001 *
Diabetes mellitus (%)	29 (31)	6 (20)	7 (35)	0.211
FM (%)	34 ± 8.8	37 ± 7.6	38 ± 7.7	0.54
HbA1c (%)	6.2 ± 1.1	5.9 ± 0.7	6.0 ± 1.2	0.65
CRP (mg/dL)	27 ± 54	19 ± 29	59 ± 47	0.04 *
Vitamin D (ng/mL)	21 ± 8.9	20 ± 10	18 ± 9.2	0.73

Data are expressed as mean ± standard deviations. Groups were divided according to vitamin D level. Abbreviations: ICU: intensive care unit; FM: fat mass; CRP: C-reactive protein; HbA1c: glycated hemoglobin. Asterisk indicates significant difference between groups, according to Student’s *t*-test (or Mann–Whitney test) (* *p* < 0.05).

**Table 10 nutrients-17-00110-t010:** Incremental changes in morphofunctional parameters according to vitamin D level in patients with obesity vs. sarcopenic obesity.

	Vitamin D < 30 ng/mL Nonsarcopenic Obesity	Vitamin D < 30 ng/mL Sarcopenic Obesity	*p* Value	Vitamin D ≥ 30 ng/mL Nonsarcopenic Obesity	Vitamin D ≥ 30 ng/mL Sarcopenic Obesity	*p* Value
	N = 16	N = 12		N = 14	N = 8	
ΔPhA	0.7 ± 0.4	0.6 ± 0.3	0.49	0.5 ± 0.2	0.8 ± 0.2	0.46
ΔSPhA	0.7 ± 0.2	0.7 ± 0.3	0.37	0.4 ± 0.1	0.5 ± 0.1	0.39
ΔTBW/FFM (%)	−0.7 ± 3.7	−0.2 ± 3.3	0.58	−1.9 ± 3.1	−2.8 ± 5.1	0.16
ΔFM (%)	−2.7 ± 1.9	−2.7 ± 1.2	0.56	−5.4 ± 0.8	−4.5 ± 0.7	0.48
ΔFFM (kg)	6.1 ± 3.1	5.8 ± 2.6	0.53	1.3 ± 0.2	2.8 ± 0.2	0.51
ΔFFMI (kg/m^2^)	2.1 ± 1.4	2.1 ± 1.5	0.67	0.4 ± 0.0.1	1.1 ± 0.1	0.45
ΔBCM (kg)	5.6 ± 2.2	3.6 ± 3.2	0.66	2.8 ± 0.3	4.3 ± 0.4	0.43
ΔBCM/h (kg/m)	3.2 ± 1.9	2.3 ± 6.37	0.59	1.6 ± 0.1	2.6 ± 0.2	0.47
ΔNa/K ratio	−0.01 ± 0.1	−0.02 ± 0.1	0.76	−0.1 ± 0.1	−0.1 ± 0.2	0.34
ΔASMM (kg)	2.1 ± 1.3	2.9± 1.4	0.66	0.1 ± 0.1	0.4 ± 0.7	0.74
ΔSMM (kg)	3.5 ± 2.9	4.2 ± 3.1	0.76	0.1 ± 0.3	0.2 ± 0.2	0.69
ΔSMI (kg/m^2^)	1.2 ± 1.2	1.6 ± 1.5	0.65	1.2 ± 1.1	1.6 ± 0.9	0.60
ΔSMM/w	3.9 ± 1.9	3.2 ± 2.1	0.68	0.03 ± 0.3	0.1 ± 0.3	0.66
ΔR-HGS (kg)	2.7 ± 2.5	7.1 ± 5.1	0.048 *	2.5 ± 9.8	8.7 ± 13	0.04 *
ΔL-HGS (kg)	1.3 ± 1.2	6.1 ± 4.0	0.07	0.5 ± 3.5	9.0 ± 9.8	0.002 *
ΔUAG (s)	−0.4 ± 0.1	−0.4 ± 1.3	0.87	−1.1 ± 1.1	−2.2 ± 1.1	0.04 *
Δalbumin (mg/dL)	1.0 ± 0.5	1.1 ± 0.6	087	0.7 ± 0.2	1.0 ± 1.1	0.43
Δprealbumin (g/dL)	9.9 ± 11	14 ± 7.6	0.40	−1.4 ± 12	4.3 ±11	0.45
ΔCRP (mg/dL)	−15 ± 14	−79 ± 61	0.20	−7.4 ± 17	−48 ± 14	0.48
ΔHbA1c (%)	−0.5 ± 0.2	0.5 ± 0.5	0.49	−0.1 ± 0.6	0.6 ± 0.7	0.29

Data are expressed as mean ± standard deviations. Groups were divided according to the time of evaluation. Abbreviations: PhA: phase angle; SPhA: standardized phase angle; TBW: total body water; FFM: fat-free mass; BCM: body cell mass; BCM/h: Sstandardized body cell mass; ASMM: appendicular skeletal muscle mass; SMM: skeletal muscle mass; SMM/w: skeletal muscle mass/weight; SMI: skeletal mass index; FM: fat mass; CRP: C-reactive protein; HbA1c: glycated hemoglobin. Asterisk indicates significant difference between groups, according to Student’s *t*-test (or Mann–Whitney test) (* *p* < 0.05).

**Table 11 nutrients-17-00110-t011:** Multivariate linear regression analysis of predictors of ΔR-HGS in patients with obesity.

Predictor	Estimate	SEE (kg)	*p*
Intercept	15.4314	6.3392	0.025
Δvitamin D (ng/mL)	0.3093	0.1276	0.025
ICU Stay (days)	0.1312	0.0696	0.319
Hospital stay (days)	0.0936	0.0485	0.048
ΔCRP (mg/dL)	−0.2105	0.1429	0.031
Diabetes mellitus (no: 0; yes: 1)	5.4654	3.8171	0.141
SMM/kg	0.1493	0.2927	0.087
Age (year)	0.0505	0.1686	0.768
Sex (male: 0; female: 1)	−3.1916	3.5959	0.914

Dependent variable: ΔR-HGS. R (multiple correlation) = 0.678; R^2^ (coefficient of determination) = 0.460. A significance level of *p* < 0.05 was considered for all two-tailed tests. Abbreviations: SEE: standard error of estimation; ICU: intensive unit care; CRP: C-reactive protein; SMM: skeletal muscle mass.

## Data Availability

Data described in this manuscript, code book, and analytic code will be made available upon request from the corresponding author.

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
