# Peer review of "Relationship Between Vitamin D Levels with In-Hospital Complications and Morphofunctional Recovery in a Cohort of Patients After Severe COVID-19 Across Different Obesity Phenotypes"

_nutrients, 2024, doi:10.3390/nu17010110_

Round 1
Reviewer 1 Report
Comments and Suggestions for Authors
Interesting study with reasonable results.
|
Evidence sug- |
119 |
|
gests that low vitamin D levels exacerbate insulin resistance and chronic low-grade in- |
120 |
|
flammation, which are central to the pathophysiology of type 2 diabetes. |
Comment: Please provide a reference with diabetes in the title for this statement.
I suggest searching a Google Scholar. It should be a reference from the past 2-3 years.
Please define FM in the text in addition to in the abstract.
|
follow-up [19,21,22] |
20 is missing
|
(Ensure® Plus Advance, Abbott Nutrition, Spain) |
Comment: Products should also include the city.
Please define BMI Grades 1-3
|
CPR (mg/dL) |
26.6±54.0 |
Should be
|
CRP (mg/dL) |
27±54 |
Figure 2. Alluvial diagram illustrating the flow of changes in vitamin D levels from 423 baseline to six months
Please define LL to HH in the figure caption. Readers should not have to search for the definitions.
Note that vitamin D levels fall in acute respiratory tract infections. Here is one reference. A better one could probably be found
Smolders, J.; van den Ouweland, J.; Geven, C.; Pickkers, P.; Kox, M. Letter to the Editor: Vitamin D deficiency in COVID-19: Mixing up cause and consequence. Metabolism 2021, 115, 154434.
Thus, it is not surprising that vitamin D levels changed over six months. Also, was the season of measurement considered?
Significant digits. The general rule is that no more non-zero digits should be given than are justified by the uncertainty of the value.
See "Too many digits: the presentation of numerical data"
https://www.ncbi.nlm.nih.gov/pmc/articles/PMC4483789/
If the uncertainty (or difference when comparing numbers) is greater than about 7%, only two non-zero digits are justified.
P values should be given to two decimal places unless the first two are 00 or the number lies between 0.045 and 0.054. If the first two are 00, then only one non-zero digit can be given.
Thus, p values should be revised.
|
61.6±11.9 should be 62±12
|
Percentages should be in whole numbers
Please review all numbers in abstract, text, tables, and figures and adjust accordingly.
Author Response
Dear Editor,
We greatly appreciate the opportunity to revise our abstract titled “Relationship between Vitamin D Levels with in-Hospital Complications and Morphofunctional Recovery in a cohort of Post-Critical COVID-19 Patients: Across Different Obesity Phenotypes”. The manuscript ID is nutrients-3384211. We thank you for the valuable feedback. Below, we provide a detailed response to each of the comments and outline the changes implemented in the manuscript:
Comment 1: Please provide a reference with diabetes in the title for this statement. I suggest searching a Google Scholar. It should be a reference from the past 2-3 years.
Response 1: added in references (reference number 17): Cojic, M., Kocic, R., Klisic, A., & Kocic, G. (2021). The effects of vitamin D supplementation on metabolic and oxidative stress markers in patients with type 2 diabetes: A 6-month follow up randomized controlled study. Frontiers in Endocrinology, 12. https://doi.org/10.3389/fendo.2021.610893).
Comment 2: Please define FM in the text in addition to in the abstract.
Response 2: Defined in the text. Line 134.
Comment 3: Reference 20 is missing.
Response 3: Added in the text. Line 149.
Comment 4: (Ensure® Plus Advance, Abbott Nutrition, Spain) Comment: Products should also include the city.
Response 4: Added in the text. Line 191.
Comment 5: Please define BMI Grades 1-3.
Response 5: Defined in Table 1.
Comment 6: CPR should be CRP.
Response 6: Modified in the entire manuscript.
Comment 7: Figure 2. Alluvial diagram illustrating the flow of changes in vitamin D levels from 423 baseline to six months. Please define LL to HH in the figure caption. Readers should not have to search for the definitions.
Response 7: Modified in Figure 3 (page 12). And we have deleted the abbreviations in the image caption.
Comment 8: Note that vitamin D levels fall in acute respiratory tract infections. Here is one reference. A better one could probably be found. Smolders, J.; van den Ouweland, J.; Geven, C.; Pickkers, P.; Kox, M. Letter to the Editor: Vitamin D deficiency in COVID-19: Mixing up cause and consequence. Metabolism 2021, 115, 154434.
Response 8: added in references (reference number 32).
Comment 9: Significant digits. The general rule is that no more non-zero digits should be given than are justified by the uncertainty of the value. See "Too many digits: the presentation of numerical data"
Response 9: Modified in the entire manuscript according to: https://www.ncbi.nlm.nih.gov/pmc/articles/PMC4483789/
Comment 10: P values should be given to two decimal places unless the first two are 00 or the number lies between 0.045 and 0.054. If the first two are 00, then only one non-zero digit can be given.
Response 10: All p values have been modified according to previous indications.
Comment 11: Percentages should be in whole numbers.
Response 11: Modified in the entire text and in tables.
Comment 12: Please review all numbers in abstract, text, tables, and figures and adjust accordingly.
Response 12: all numbers have been reviewed and modified.
The changes are written in red in the new draft of the manuscript.
Thank you again for your constructive feedback and for considering our submission for publication.
Sincerely,
Víctor J. Simón Frapolli.
Reviewer 2 Report
Comments and Suggestions for Authors
This study investigates the relationship between vitamin D levels and in-hospital complications, as well as morphofunctional recovery, in post-critical COVID-19 patients. The authors emphasize the importance of vitamin D in influencing both the severity of illness and the recovery process, especially in individuals with obesity and sarcopenia. The study's findings indicate that vitamin D deficiency is associated with longer hospital and ICU stays, higher levels of inflammation, and worse muscle health, while normal vitamin D levels are linked to improved recovery outcomes. The sample size of 94 post-critical COVID-19 outpatients provides a relatively well-defined cohort for examining the relationship between vitamin D and recovery in a clinical setting. The study also evaluates a wide range of clinically relevant outcomes, including in-hospital complications (e.g., ICU stay, mechanical ventilation), muscle mass and strength (e.g., handgrip strength, skeletal muscle mass), and markers of systemic inflammation (e.g., CRP, HbA1c). These measures give a thorough understanding of how vitamin D levels influence both short- and long-term recovery. The inclusion of sarcopenic and non-sarcopenic obese patients adds depth to the analysis and highlights the unique recovery challenges faced by this subgroup, which is important for personalized treatment approaches. Additionally, the six-month follow-up provides valuable insights into the longer-term effects of vitamin D levels on recovery, strengthening the study’s relevance to post-critical care management.
However, the study's observational design limits its ability to establish causality. While it identifies correlations between vitamin D levels and various outcomes, future research using randomized controlled trials (RCTs) would provide more robust evidence regarding the effectiveness of vitamin D supplementation in improving recovery outcomes for COVID-19 patients. The study does not provide detailed information about whether patients with deficient or insufficient vitamin D levels were given supplementation during their hospitalization or follow-up period. Including this data could clarify whether improvements in outcomes are directly linked to vitamin D supplementation or merely to baseline levels. Furthermore, while the study mentions "adapted nutritional support" and "specific physical rehabilitation," these interventions are not described in detail. Understanding how these factors interact with vitamin D levels would provide greater insight into the potential synergistic effects of combined therapies in improving functional recovery. Additionally, the study could benefit from a more in-depth analysis of the specific components of the rehabilitation program, including the intensity, duration, and type of physical activity, to assess how these interact with vitamin D levels.
While the study looks at sarcopenic versus non-sarcopenic obesity, a further breakdown of these groups by other factors such as age, gender, and pre-existing conditions (e.g., diabetes, hypertension) could reveal more nuanced insights into which subgroups benefit most from maintaining normal vitamin D levels. The study also suggests that the impact of vitamin D on muscle recovery was greater in sarcopenic obese patients, but it would be useful to explore why this subgroup responds more favorably, particularly in relation to their specific metabolic or inflammatory profiles. The statistical methods used to control for potential confounders, such as age, gender, comorbidities, and baseline vitamin D levels, should be more clearly outlined. While the study presents significant findings, a more detailed explanation of the multivariable regression models and how they accounted for confounders would increase the transparency and rigor of the results. Moreover, the study primarily focuses on patients with obesity and sarcopenia, which limits its generalizability to a broader population. Including patients without these conditions could allow for a more comprehensive understanding of the relationship between vitamin D and recovery across different health statuses.
Future research should focus on randomized controlled trials to more definitively assess the causal relationship between vitamin D supplementation and improved outcomes in post-critical COVID-19 patients. Extending the follow-up period beyond six months would help assess whether the benefits of normal vitamin D levels persist over time and whether long-term supplementation is necessary to maintain these benefits. Furthermore, mechanistic studies examining the underlying processes by which vitamin D influences inflammation, muscle mass, and strength could provide deeper insights into how this micronutrient contributes to recovery in post-critical COVID-19 patients. The study provides valuable insights into the role of vitamin D in recovery from COVID-19, particularly in obese and sarcopenic patients. It underscores the importance of maintaining normal vitamin D levels to optimize recovery outcomes. However, the observational design and lack of detailed data on supplementation and rehabilitation strategies limit the conclusions that can be drawn from the results. Future studies, particularly randomized controlled trials, are needed to confirm the findings and explore the optimal strategies for improving recovery in this vulnerable patient population.
Author Response
Dear revisor,
We greatly appreciate the opportunity to revise our abstract titled “Relationship between Vitamin D Levels with in-Hospital Complications and Morphofunctional Recovery in a cohort of Post-Critical COVID-19 Patients: Across Different Obesity Phenotypes”. The manuscript ID is nutrients-3384211. We thank you for the valuable feedback. Below, we provide a detailed response to each of the comments and outline the changes implemented in the manuscript:
Comment 1: While it identifies correlations between vitamin D levels and various outcomes, future research using randomized controlled trials (RCTs) would provide more robust evidence regarding the effectiveness of vitamin D supplementation in improving recovery outcomes for COVID-19 patients.
Response 1: Detailed in "Limitations" (698-709): Regarding in-hospital complications analysis, another important limitation is that our results demonstrate association rather than causality, given that vitamin D levels were assessed after admission and not prior to it. Future research using randomized controlled trials (RCTs) would provide more robust evidence regarding the effectiveness of vitamin D supplementation in improving recovery outcomes for COVID-19 patients and the optimal strategies for achieve it in this vulnerable patient population. Finally, extending the follow-up period beyond six months would help assess whether the benefits of normal vitamin D levels persist over time and whether long-term supplementation is necessary to maintain these benefits. Furthermore, mechanistic studies examining the underlying processes by which vitamin D influences inflammation, muscle mass, and strength could provide deeper insights into how this micronutrient contributes to recovery in post-critical COVID-19 patients.
Comment 2: The study does not provide detailed information about whether patients with deficient or insufficient vitamin D levels were given supplementation during their hospitalization or follow-up period. Including this data could clarify whether improvements in outcomes are directly linked to vitamin D supplementation or merely to baseline levels.
Response 2: Detailed in Methods (lines 203-208): Additionally, patients with vitamin D deficiency or insufficiency received oral 25-hydroxycholecalciferol supplementation during follow-up, in accordance with clinical practice guideline recommendations: if vitamin D insufficiency, 1500-2000 IU/day until normalization. If vitamin D deficiency, 50,000 IU/week for 8 weeks, until vitamin D levels greater than 30 ng/ml were reached, followed by maintenance with doses of 1500-2000 IU/day [13,25].
Comment 3: Furthermore, while the study mentions "adapted nutritional support" and "specific physical rehabilitation," these interventions are not described in detail. Understanding how these factors interact with vitamin D levels would provide greater insight into the potential synergistic effects of combined therapies in improving functional recovery. Additionally, the study could benefit from a more in-depth analysis of the specific components of the rehabilitation program, including the intensity, duration, and type of physical activity, to assess how these interact with vitamin D levels.
Response 3: Detailed in Methods (lines 187-191 for "adapted nutritional support" and lines 193-202 for "specific physical rehabilitation"): "...and were instructed to follow a mediterranean patern oral protein-enriched diet and to supplement it with two servings of an Oral Nutritional Supplement (ONS) daily. The ONS provided 1.5 kcal/mL, delivering 330 kcal, 20 g protein, 11 g fat, 37 g carbohydrates, 1.7 g fiber, 1.5 g calcium HMB, and 500 IU vitamin D per 220 mL serving (Ensure® Plus Advance, Abbott Nutrition, Madrid, Spain)" and "In this context, once the patients had been assessed with global joint balance and global muscle balance in Rehabilitation consultation and if they presented any type of deficit, they were indicated the possibility of carrying out a comprehensive rehabilitation program, on-site or at home (Figure S1), depending on the following characteristics. The indications for an on-site rehabilitation program were: COVID-19 patients with negative PCR test, with capacity for independent ambulation, stable, presenting dyspnea of moderate to great effort and/or fatigue with moderate effort, fragile with SPPB<10 and/or need for O2. The indications for a home rehabilitation program were: fragile patients with no need for O2, patients who could not attend a face-to-face rehabilitation program".
Comment 4: While the study looks at sarcopenic versus non-sarcopenic obesity, a further breakdown of these groups by other factors such as age, gender, and pre-existing conditions (e.g., diabetes, hypertension) could reveal more nuanced insights into which subgroups benefit most from maintaining normal vitamin D levels. The study also suggests that the impact of vitamin D on muscle recovery was greater in sarcopenic obese patients, but it would be useful to explore why this subgroup responds more favorably, particularly in relation to their specific metabolic or inflammatory profiles. The statistical methods used to control for potential confounders, such as age, gender, comorbidities, and baseline vitamin D levels, should be more clearly outlined. While the study presents significant findings, a more detailed explanation of the multivariable regression models and how they accounted for confounders would increase the transparency and rigor of the results. Moreover, the study primarily focuses on patients with obesity and sarcopenia, which limits its generalizability to a broader population. Including patients without these conditions could allow for a more comprehensive understanding of the relationship between vitamin D and recovery across different health statuses.
Response 4: Thank you for this appreciation. It has allowed us to perform this subanalysis and draw further conclusions. We have performed a multivariate linear regression analysis, and we have detected that in addition to the increase in vitamin D, longer hospital stay and a greater decrease in CRP are associated with statistical significance to the improvement in R-HGS in follow-up. Nevertheless, the increase in vitamin D continues to have an impact independently of the previous variables, but we have been able to draw the conclusion that these results suggest that patients with sarcopenic obesity improve more than those with non-sarcopenic obesity not only because of the effect of vitamin D itself, but also because patients with sarcopenic obesity partly present a higher degree of inflammation as a result of a longer hospital stay, which also impacts on the recovery of R-HGS. Exposed on lines 42-44, 405-504, 656-667, 715-716, in red color.
Comment 5: Future research should focus on randomized controlled trials to more definitively assess the causal relationship between vitamin D supplementation and improved outcomes in post-critical COVID-19 patients. Extending the follow-up period beyond six months would help assess whether the benefits of normal vitamin D levels persist over time and whether long-term supplementation is necessary to maintain these benefits. Furthermore, mechanistic studies examining the underlying processes by which vitamin D influences inflammation, muscle mass, and strength could provide deeper insights into how this micronutrient contributes to recovery in post-critical COVID-19 patients. The study provides valuable insights into the role of vitamin D in recovery from COVID-19, particularly in obese and sarcopenic patients. It underscores the importance of maintaining normal vitamin D levels to optimize recovery outcomes. However, the observational design and lack of detailed data on supplementation and rehabilitation strategies limit the conclusions that can be drawn from the results. Future studies, particularly randomized controlled trials, are needed to confirm the findings and explore the optimal strategies for improving recovery in this vulnerable patient population.
Response 5: Since it is a topic related to comment 1, we believe that answer 1 would be also valid to answer it.
Thank you again for your constructive feedback and for considering our submission for publication.
Sincerely,
Víctor J. Simón Frapolli.